# Collision-Aware Routing Using Multi-Objective Seagull Optimization Algorithm for WSN-Based IoT

**DOI:** 10.3390/s21248496

**Published:** 2021-12-20

**Authors:** Preetha Jagannathan, Sasikumar Gurumoorthy, Andrzej Stateczny, Parameshachari Bidare Divakarachar, Jewel Sengupta

**Affiliations:** 1Department of Computer Science and Engineering, Muthayammal Engineering College Autonomous, Rasipuram 637408, Tamil Nadu, India; preetha.j.cse@mec.edu.in; 2Department of Computer Science and Engineering, Jerusalem Collège of Engineering, Chennai 600100, Tamil Nadu, India; drgsasikumar@veltech.edu.in; 3Department of Geodesy, Gdansk University of Technology, 80-233 Gdansk, Poland; 4Department of Telecommunication Engineering, GSSS Institute of Engineering and Technology for Women, Mysuru 570016, Karnataka, India; paramesh@gsss.edu.in; 5Department of Informatics Engineering, Kaunas University of Technology, 44249 Kaunas, Lithuania; jewel.sengupta@ktu.edu

**Keywords:** congestion, Internet of Things, scalability, seagull optimization algorithm, wireless sensor network

## Abstract

In recent trends, wireless sensor networks (WSNs) have become popular because of their cost, simple structure, reliability, and developments in the communication field. The Internet of Things (IoT) refers to the interconnection of everyday objects and sharing of information through the Internet. Congestion in networks leads to transmission delays and packet loss and causes wastage of time and energy on recovery. The routing protocols are adaptive to the congestion status of the network, which can greatly improve the network performance. In this research, collision-aware routing using the multi-objective seagull optimization algorithm (CAR-MOSOA) is designed to meet the efficiency of a scalable WSN. The proposed protocol exploits the clustering process to choose cluster heads to transfer the data from source to endpoint, thus forming a scalable network, and improves the performance of the CAR-MOSOA protocol. The proposed CAR-MOSOA is simulated and examined using the NS-2.34 simulator due to its modularity and inexpensiveness. The results of the CAR-MOSOA are comprehensively investigated with existing algorithms such as fully distributed energy-aware multi-level (FDEAM) routing, energy-efficient optimal multi-path routing protocol (EOMR), tunicate swarm grey wolf optimization (TSGWO), and CoAP simple congestion control/advanced (CoCoA). The simulation results of the proposed CAR-MOSOA for 400 nodes are as follows: energy consumption, 33 J; end-to-end delay, 29 s; packet delivery ratio, 95%; and network lifetime, 973 s, which are improved compared to the FDEAM, EOMR, TSGWO, and CoCoA.

## 1. Introduction

The WSN is a self-organizing communication system that comprises terminal structures, sensor nodes, and base stations (BSs). The IoT represents innovative thinking and progresses numerous communication structures, which depend upon WSNs for collecting the data, and it can be improved through cloud/edge/fog computing [1,2,3,4]. In this way, several services and applications of the IoT have been presented to transfer the data packets. The IoT plans to create widespread Internet, which could influence different portions of consumers’ requirements [5,6]. The exploitation of energy for the duration of the routing procedure might cause a problem because of inadequate sensor strategies. Numerous nodes are transferred in the land section over a few wireless types of machinery to analyze the extensive possibility [7]. In IoT, WSN is a fundamental constituent that has increased into more than a few dissimilar practical presentations [8]. Nowadays, WSNs and IoT have been used in numerous risky and non-critical applications, which motivated to increase their life expectancy [9]. WSN nodes are insignificant in battery-driven technologies [10], so, the energy-efficient method raises the network lifespan, which is exceedingly important. Till now, several methodologies and procedures have been used for energy-efficient concepts in WSN-based IoT structures [11]. Moreover, the packet loss and higher delay times occur because of congestion during the communication. Since, the congestion occurs because of the restricted channel capacities [12,13].

In the present situation, not only have WSNs developed as important equipment, but they are also recognized as playing a vital role in numerous applications, such as the IoT [14]. However, the WSN resources are inadequate, therefore the routing scheme turns out to be complex. For that reason, there is a need to create an enhanced routing protocol that deliberates the energy-related issues [15,16]. Many types of research focus on the clustering and routing design and additional schemes for energy conservation [17]. In the current situation, quality of service (QoS) must be enhanced because the system should continue over an extended time [18]. After this process, additional information about a structure is given through the optimization method [19]. The inadequacy of WSN resources generate the designing challenge of an energy-efficient transmitting route and network configuration [20]. Generally, these ideas have not appeared for any security and these properties should be utilized sensitively [21]. This present research concentrates on clustering/routing protocols established for WSN-dependent IoT applications. Here, the routing procedure plays an essential part while transmitting the collected data from sensors to the endpoint for additional control. The processed data could be exploited in several applications for assisting individuals to advance their way of life [22,23].

The paper presents CAR-MOSOA for enhancing the lifetime of a network using optimization-based clustering and routing in WSN-based IoT. The MOSOA is one of the metaheuristic algorithms that imitate the natural behavior of the seagulls [24,25].

The major contribution of this research is as follows:A CAR-MOSOA based cluster head (CH) with efficient fitness function is developed. This seagull optimization algorithm (SOA) is selected because it has the collision avoidance property in nature, so it is mainly helpful to generate collision-aware routing in the network.A CAR-MOSOA-based routing algorithm is developed with a trade-off between transmission distance and a number of data that travel with an efficient particle encoding scheme for the routing process.A simulation of the CAR-MOSOA is analyzed to demonstrate the superiority over other existing algorithms.

The organization of this research paper is given as follows: The survey of the recent techniques related to clustering and routing in WSN-IoT is described in Section 2. The problem statement of this research is given in Section 3. The preliminaries and their system model are explained in Section 4. The description of the proposed CAR-MOSOA method with a block diagram is specified in Section 5. The simulation results and its comparative analysis of the CAR-MOSOA method are declared in Section 6. Finally, the conclusion part is stated in Section 7.

## 2. Related Work

There are numerous existing procedures, methods, and ideas from conventional wireless networks that are yet to be exploited in WSN-IoT; however, several important alterations lead to the necessity of innovative procedures and methods. In this segment, some of the collected works concerning the clustering and routing in WSN-IoT are deliberated.

Jennifer S. Raj, Dr. Abul Basar [26] projected an energy efficient model through fuzzy rule and neural networks clustering and routing (FNCR) for attaining the quality of service (QOS) factors. The projected procedure employs the sequence of clustering with FNCR, which was maintained through the shortest route with improved routing competencies for WSN-IoT. Furthermore, the proposed FNCR consumed less energy and managed the QOS performance metrics. However, the FNCR has high energy consumption when compared with other machine learning techniques.

Maryam Shafiq et al. [27] demonstrated the robust cluster-based routing protocol (RCBRP) for WSN-IOT to recognize routing in which a smaller amount of energy was expended to improve the life expectancy of a network. This suggested system offered various stages to discover the transmission. Here, the energy-efficient clustering/routing process, distance, and energy depletion control procedure have been analyzed. The analysis clearly shows that the recommended method consumes less energy and stabilizes the load by gathering the devices. However, at certain rounds, only a few nodes are alive, so that the functioning of the network gets discontinued, and it reaches a dangerous situation.

Nalluri Prophess, Raj Kumar, Josemin Bala Gnanadhas [28] presented the cluster centroid-based energy-efficient routing (CEER) procedure for WSN-supported IoT to enhance the lifespan of the network and minimize the energy depletion. The projected CEER was exploited to resolve the issue of establishing zones to transfer the data professionally to the base station (BS). Moreover, the routing process deliberates the nodes’ residual energy and distance to BS. If the topology gets varied, the position of the node also varies and moves beyond its sensing level.

Mohammad Ali Alharbi, Mario Kolberg, and Muhammad Zeeshan [29] demonstrated an energy-efficient clustering/routing resolution for hot-spot difficulties present in WSN-IoT. This clustering procedure depends on the position of nodes and exploits a CH procedure, which chooses a CH that controls the node connectivity. Correspondingly, this proposed routing procedure manages weak situations using marginal paths to designate the cluster leader. Conversely, it attains the better throughput standards for a few communication nodes only, later it fails to produce the best throughput. 

Ipek Abastkeles-Turgut Gokhan Altan [30] presented a clustering and routing procedure called fully distributed energy-aware multi-level (FDEAM) for WSN-IoT. Here, two-level/multi-level inter-cluster transmission processes are projected. Moreover, the exposure region of second-level grouping was enthusiastically formed conferring to cluster distance to BS. Self-arranged nodes elect the limits for clustering, and cluster heads are selected by executing FDEAM method. Despite this, this method was not appropriate for non-uniform node allocations and had the shortcoming of being reliant on a dominant source.

Kavita Jaiswal and Veena Anand [31] demonstrated an energy-efficient optimal multi-path routing protocol (EOMR) to enhance the QoS metrics in WSN-IoT presentations. This EOMR protocol allows the source node to select a route for data transmission from the set of identified multiple routes. Furthermore, it deliberates multiple aspects to choose the ideal route for improving the lifespan and traffic amount at the subsequent nodes. The transmission processes happen regularly in definite portions of the network, although some portions are not utilized regularly throughout the period.

Nitesh Chouhan and S. C. Jain [32] presented the multi-path routing protocol using the technique termed the tunicate swarm grey wolf optimization (TSGWO) procedure in the WSN-IoT. By using TSGWO, the multiple routes were calculated by the source node to numerous endpoints. The source node promotes the data packet to the end point instantaneously. On the other hand, the route with negligible delay, routing distance, and highest linkage lifespan was selected as the ideal route. However, this TSGWO procedure does not consider the maintenance parameters for the calculation.

Betzler et al. [33] developed the constrained application protocol (CoAP) for the IoT. Subsequently, this CoAP simple congestion control/advanced (CoCoA) was developed for improving the performances of the IoT. This CoCoA was combined with the utilization of retransmission timeout (RTO) aging approaches, dynamic RTO backoff measurements, and round-trip time (RTT) measurements for achieving the evaluation of dynamic RTO to forward the CoAP messages. The congestion control offered by CoCoA was convenient for IoT features and flexible to network dynamics. However, this CoCoA only provided congestion control for the IoT.

From the literature review, it was observed that the WSN-IoT network paradigm is the recent trend that attracts researchers due to its vulnerability. For over a decade, the research community is working toward developing energy-efficient routing mechanisms to increase network longevity. The proposed protocols with a detailed description of their methodologies and focus on the issues that lead to high energy consumption in the nodes with different activities are discussed in the following section.

## 3. Problem Statement

The main challenge in energy-efficient routing is to have data transmission without connectivity by exploiting stringent controlling procedures.Usually, sensor nodes depend on battery energy, which gets exhausted at a quicker rate due to calculation and transmission processes [34].The single-hop transmission between the CH and BS creates a high energy consumption over the network [35]. Therefore, data reliability must be preserved during the data packets’ transmission.In WSN, the progressing of multiple subjects and node energy become serious issues because of inadequate power delivered to the nodes [36].

Solution:

To obtain a faster and efficient solution for the clustering and routing problem, a metaheuristic approach or classical approach is exceedingly required. Therefore, this research paper developed CAR-MOSOA-based clustering and routing algorithms for WSN-IoT with the consideration of energy consumption of the sensor nodes for prolonging network lifetime. In this CAR-MOSOA, optimal CHs are selected from the clustering by considering distinct fitness functions such as network coverage, communication cost, residual energy, and node degree. This optimal CH selection is used to achieve load balancing between the nodes and to minimize the energy consumption. Moreover, the routing over the network is accomplished using the queue length, link quality, communication cost, and residual energy. Here, the queue length considered in the MOSOA leads us to achieve congestion-aware routing over the WSN-IoT.

## 4. Preliminaries

The main objective of this research is to generate collision-free (i.e., congestion-aware) and energy-efficient routing for transmitting the emergency messages over the WSN-IoT and also transmit the alert messages using the IoT devices to the respective destination with a minimum delay. A multi-tier hierarchical framework based on subdivision technique for node deployment that aims to reduce long-distance communication. The proposed multi-tier framework can support load balancing and scalability. Scalability is a significant aspect in planning a well-organized routing protocol for WSN-IoT. A decent transmitting procedure has to be accessible and adaptive to alterations in the network configuration.

### 4.1. Multi-tier Framework

Figure 1 represents the multi-tier dependent clustering architecture. The nodes present in this structure are either mobile or static, which depends upon the necessity of the application. The concept of node mobility was conducted by developing the random walk (RW) and random waypoint (RWP) model. 

In this architecture, the quadrangular profile system zone is deliberated. The upper level is constructed as the BS, and the system zone is separated into several regions. Each region is distributed into sub-regions, then it is represented as a cluster. The regions are condensed in the direction of the BS, and cluster counts are enlarged in the direction of BS to steady the system load equivalently [37]. The lower level has the determined extent through the lowest clusters count, and the upper level has the lowest extent through the higher cluster count. The motive enumerated in this research is that the upper region not only transfers collected information toward the BS but also relays the traffic of lower regions. In additional confrontations, the intra-cluster distance is condensed as moving on the road to BS via cluster count through the diminishing region. The system model of WSN-IoT is explained in the following section.

### 4.2. System Model

Commonly, the WSN comprises a high quantity of low-priced and insignificant sensor nodes. These nodes are frequently utilized in dissimilar presentations, for example, environmental observation, gravity, moisture, estimating meteorological conditions, detecting volcanic tremors, vehicular activities, and army investigation. Furthermore, the node with substantial load traffic is deliberated as the chief influence that should be resolved to permit improvement in the features of WSN. The network model comprehends a sensor node through a sink node that is commonly positioned at the center of the recognition zone. The network model with WSN-IoT is illustrated in Figure 2. 

In this research, CAR-MOSOA is considered to decrease the consumption of energy in the WSN. In the WSN, the dead nodes correspondingly expended energy to direct the information; however, it decreases the data speed and consumes additional energy from the series, which diminishes the presentation and lifespan of the network. To eliminate this, CAR-MOSOA-dependent clustering process is presented, which recognizes the finest node for each clustering stage. Subsequently, the node in the system with high energy requirements need to be designated as the sink node that is termed as the leader of the cluster group. This is because the leader recompenses the main part in the structure to eliminate the additional energy depletion that qualifies the appropriate transmission practice in the network.

### 4.3. Energy Model

Here, the energy model is calculated and the consumption of energy in the WSN arrangement is determined. The structural design of the WSN is particularly planned through dualistic categories of communication networks, for example, multi-path channels depending on the distance between receiver, transmitter, and free space [38,39]. Once the multi-path channel is employed, the channel distance is continually lessened over the sample free space model. The architecture of the energy model for source/destination is arithmetically expressed in Equation (1),
(1)$$Ener^{T}\left(1,D\right)=\left\{\begin{array}{c} L\times E^{Elec}+1*\;\in^{FS}\times D^{2},\;D<D^{0}\\ L\times E^{Elec}+1*\times D^{2},\;D\geq D^{0} \end{array}\right\}$$
where the energy required by multi-path channels is characterized as $\epsilon^{MP}$, and the energy needed for free space is specified as $\epsilon^{FS}$. Moreover, the energy needed in place of an electrical circuit is designated as $E^{Elec}$. The overall energy required by the transmitter through the distance, d, is calculated in Equation (2). The energy prerequisite of a receiver is expressed as Equation (2),
(2)$$Ener^{R}\left(L\right)=L\times E^{Elec}$$
where the receiver’s essential energy is specified as *En*(*L*), the distance between transmitter and receiver is stated as $\epsilon^{MP}\times D^{2}$ and $\in^{FS}\times D^{2}$. $E^{Elec}$ is calculated through numerous features, for example, signal filtering, spreading, alphanumeric coding, and variation.

### 4.4. Overview of Seagull Optimization Algorithm

Here, the SOA is established to choose the leader of cluster/routing progression. A comprehensive explanation of SOA is obtainable in this segment. Seagulls belong to the family Laridae and can be found in the wide-ranging biosphere. There are numerous types of sea birds, but seagulls has attractive characteristics such as, they are persistent and their prey drive is strong. Therefore, the seagulls are referred to as clever birds [40] and display specific migration and hunting behaviors. Seagulls are chosen over other seawater and freshwater birds because of their extraordinary features and quick decision-making abilities [41,42]. The seagull optimization algorithm has two important processes, i.e., migration and attacking, which are explained subsequently. 

#### 4.4.1. Migration

In process of migration, the seagull needs to recompense the following multiple scenarios.

i.Collision avoidance

In the SOA, in order to avoid impact between neighbors, extra constraints are in place for the calculation of the ideal exploration agent location as expressed in Equation (3).
(3)Cs=A×Ps(X)
where the drive features of the search agent are characterized as A, present iteration is stated as X, search agent’s present location is specified as Ps, and Cs is not impacted by residual agents. The search agent’s movement patterns are obtainable in Equation (4),
(4)$$A=Fc-(X\times \left(\frac{Fc}{maxiteration}\right)$$
where X=0,1,2,…max iteration. 

Fc is designated as 2; A is labeled linearly and condensed from Fc to 0; Fc is used to regulate the constraint’s frequency.

ii.Movement in the direction of optimal neighbors

As soon as collision between neighbors is eliminated, the exploration agents are detected in the direction of optimal neighbor movements, which is formulated as (5).
(5)Ms=B×(Pb(X)−PS(X))
where the search agent and its position is specified as  PS(X) & Ms; B is labeled as a random agent, which is accountable for effectual assessment between examination and manipulation; the exploration agent with optimal fitness is stated as Pb(X); and the random value calculation is expressed in Equation (6)
(6)$$B=2\times S=A^{2}\times RD$$
where RD is designated as a random quantity accessible in diversity between [0,1].

iii.Remain close to the finest search agent

At last, the location of the updated search agent is connected to the ideal search agent, which is expressed in Equation (7),
(7)DS=|Cs+Ms|
where DS is defined as the distance between the actual search agent and the best-fit search agent.

#### 4.4.2. Attacking the Prey

The reason for the utilization of this algorithm is the reduced amount of calculation involved in the exploration process. During the attacking process, seagulls modify the migration state, in which their focus is on preserving their altitude based on air currents and weight. In the course of attacking the prey, they may perform twisting motions in midair. This twisting drive performance can be described as in Equations (8)–(11).
(8)X=R×cosK
(9)Y=R×sinK
(10)Z=R×K
(11)$$R=U\times e^{KV}$$
where actual logarithm base is specified as e; spiral shape quantities are defined as u and v; k is labeled as an indiscriminate amount by the range [0≤k≤2π]; and the extent of the spiral on each chance is signified as R. The updated progression of the search agent is calculated through Equation (12),
(12)Ps(X)=(DS×X×Y×Z)+Pbs(X)
where Pbs(X)  is designated as an optimum response and describes the position of the remaining search agents.

## 5. Proposed Method

Here, in this research, collision-aware routing using multi-objective SOA (CAR-MOSOA) is designed to decrease the additional consumption of energy at each node. At preliminary circumstances, every node is produced through comparable energy bases. For the duration of transmission, energy is condensed for every node and determined by the progression. Figure 3 shows the overall block diagram for the energy-efficient WSN-IoT.

Step 1: At first, the sensor node is deployed.Step 2: Then the multi-tier-based clustering model is created along with the sensor nodes.Step 3: After that, the process of the fitness function is calculated based on the proposed MOSOA.Step 4: Then the process of CH selection occurs and progress the routing over WSN-IoT based on MOSOA.Step 5: After the clustering/routing process, the transmitted data is analyzed (to determine whether it is collision free or not).

### 5.1. CH Selection

CH selection and routing are done using the MOSOA. The proposed MOSOA is selected because it has a collision-avoidance property by nature, so it is mainly helpful to generate congestion-aware routing during emergencies. 

#### Fitness Function

The proposed MOSOA chooses the secure optimal CHs from the clusters for obtaining secure data transmission over the network. The purpose is to choose the most optimal number of nodes such as CHs. The objective is to accomplish, proper fitness by formulating the network coverage, communication cost, residual energy, and node degree. The parameters used in the clustering optimization are as follows:

a.Network coverage

The network coverage can be defined by Equation (13),
(13)$$N_{cov}=r\left(N_{i}\right)$$
where $r\left(N_{i}\right)\;$ represents the radius covered by a node. The first objective is represented as
$$Minimize\;f_{1}=\frac{1}{N_{T}}\overset{N}{\underset{i=1}{\sum }}N_{cov}\left(N_{i}\right)$$

b.Communication cost

The cost essential for transmitting to the neighbor node is described in Equation (14).
(14)$$C_{com}=\frac{d_{avg}^{2}}{d_{0}^{2}}$$
where $d_{avg}^{2}$ is specified as the distance between the neighbor and node; node radius is defined as $d_{0}^{2}$.

The second objective is represented as
$Minimize\;f_{2}=\frac{1}{N_{T}}\overset{N}{\underset{i=1}{\sum }}N_{prox}\left(N_{i}\right)$, where *N* represents a number of nodes.

c.Residual energy

The third objective of the residual energy is $f_{1},$ which is reduced and represented in Equation (15).
(15)$$Minimize\;f_{3}=\overset{m}{\underset{i=1}{\sum }}\frac{1}{E_{CHi}}$$

d.Degree of nodes

Node degree is defined as the quantity of non-CH participants that go to the particular mobile node. If a cluster head formerly had reduced participants, it sustains for an extensive period, due to a preference for the lesser degree of nodes. Hence, the final objective is $f_{3},$ which is decreased in Equation (16).
(16)$$Minimize\;f_{4}=\overset{m}{\underset{i=1}{\sum }}I_{i}\;$$

The above-declared objectives are convert the multi-objective function into a single objective. Consequently, the normalization process (F(x)) is applied to every objective $\alpha_{1},\alpha_{2},$ $\alpha_{3},\alpha_{4}$.
(17)$$F\left(x\right)=\frac{f_{i}-f_{min}}{f_{max}-f_{min}}\;$$
where function value is signified as $f_{i}$, and $f_{min}$ and $\;f_{max}$ are specified as the minimum and maximum fitness values.
(18)$$Minimum\;fitness=\alpha_{1}f_{1}+\alpha_{2}f_{2}+\alpha_{3}f_{3}+\alpha_{4}f_{4}\;$$
where $\overset{4}{\underset{i=1}{\sum }}\alpha_{i}=1;\mathrm{and}\;\alpha_{i}\in \left(0,1\right)$.

In the process of routing, every population dimension is identical to the quantity of cluster heads (m). Let $P_{i}=(P_{i}^{1},P_{i}^{2},\ldots P_{i}^{m}$) be ith population, where each dimension of a population, i.e., $P_{i}^{1}=\left(0,1\right),$ is randomly initialized. Further, a new plotting method is employed to determine the following node in the direction of BS.

### 5.2. Routing Using MOSOA

In this phase, the collision-free optimal transmission path from the source node to the destination is discovered using the MOSOA. The routing path generation is optimized using the following fitness metrics such as queue length, link quality, communication cost, and residual energy. This path generation has various phases such as representation, initialization, fitness function derivation, and iterative process. These processes are detailed subsequently.

#### 5.2.1. Representation and Initialization of Seagull

The probable solution for the MOSOA is represented as a seagull. The population of seagulls characterizes that the transmission path contains a cluster head. In particular, the seagull has x and y coordinates, and the initialization of the MOSOA is formulated in Equation (19).
(19)$$Cs_{i}=\left\{Cs_{i,1},\;Cs_{i,2},\ldots ,Cs_{i,m}\right\}$$
where m is the number of CHs in the respective transmission path and $1\leq Cs_{i,a}\leq m$ defines the successive CH in the route.

#### 5.2.2. Fitness Function

The parameters used in the routing optimization are as follows:

a.Queue length

QL, shown in Equation (20), is considered as a primary fitness value during the routing as it considers the congestion level of each node in the WSN-IoT. Since the developed CAR-MOSOA is required to transmit the alert messages over the WSN-IoT, this QL is used to improve the data delivery performances.
(20)$$QL=\frac{RP_{k}}{Total\;buffer}$$
where the received packets at the kth node are represented as $RP_{k}$.

b.Link quality

Link quality is used to define the successful data delivery between the nodes k and l based on the amount of data packet transmissions and the retransmissions, which is expressed in Equation (21).
(21)$$Link\;quality=\frac{1}{f\times r}\;$$
where, f and r define the forward and reverse data transmission among the nodes.

The details of the communication cost and residual energy are already defined in the section. Subsequently, all the multiple objective’s fitness is conflicting with each other, so it is transformed into a single-objective fitness value as shown in Equation (22)
(22)$$Routing\;fitness=\delta_{1}\times QL+\delta_{2}\times Link\;quality+\delta_{3}\times CC+\delta_{4}\times RE$$
where, $\delta_{1},\;\delta_{2},\;\delta_{3},$ and $\delta_{4}$ are the weighted parameters, which are equal to 0.3, 0.25, 0.25, and 0.2, respectively; CC and RE represent the communication cost and residual energy, respectively.

#### 5.2.3. Iterative Process

The iterative process of the MOSOA-based CAR is as follows:

At first, the seagulls are initialized with the routing path information, and the seagull has the information about the x and y coordinates of the next-hop CHs. The best population among the entire seagulls is identified based on the fitness parameter expressed in Equation (22).Equation (22) is used to avoid collision during the migration process. After accomplishing the collision avoidance, the remaining populations are updated based on the best population (i.e., optimal routing path).Further, the positions of the seagulls are updated according to the exploitation phase. In this phase, the locations are updated by using the best solution from the MOSOA.Finally, the developed MOSOA provides the optimal routing path from the source CH to the destination node.

The data packet transmission is initialized once the routing path is identified using the MOSOA. The queue length is used to generate the collision-free path, which transmits the data over the network through the node that has less congestion. Next, the link quality is used to discover the value of the route based on the sent and received packets. The transmission distance over the WSN-IoT is minimized by considering the communication cost, and the residual energy to select the node with high remaining energy. Therefore, these fitness values are used to select the optimal collision-aware route for alert message transmission.

## 6. Results and Discussion

This section discusses the results of the developed CAR-MOSOA for multicast routing protocol in WSN-based IoT. The results and discussion of this proposed SOA method are clearly described in this section. The Network Simulator-2.34 (NS-2.34) was used for the implementation of the CAR-MOSOA, in that the system used a 4-GB RAM and Intel Core processor. The implemented SOA method was used to accomplish secure data transmission between the source to the BS. The performance of the proposed algorithm was studied based on the following parameters:Energy consumption: It is defined as the overall energy consumed by the system for the period of data packet transmission.End-to-end delay: It is defined as the period between the transmission of the data packet from the source node and its arrival at the destination node.Packet delivery ratio (PDR): It is demarcated as the proportion of the number of packets delivered to the endpoint to the number of packets transmitted by a base station.Life time: It is calculated at each node while passing through the route request. On the other hand, each node guesses the route lifespan between the actual and aforementioned nodes.

The design and simulation of the CAR-MOSOA method in NS-2.34 with various constraints such as network size, traffic type, packet size, and so on are listed in Table 1.

To analyze the effect of CBR traffic type, the network size was considered as 500 × 500. The number of nodes was fixed as 400. The proposed CAR-MOSOA was effectively executed and compared with customary cluster-based methodologies, for example, the proposed calculations and their prevalent execution such as energy consumption, end-to-end delay, PDR, the throughput of the above-mentioned node counts. Here, the existing approaches such as FDEAM [30], EOMR [31], TSGWO [32], and CoCoA [33] were implemented using the same specifications mentioned in Table 1.

### 6.1. Performance of Energy Consumption

The performance analysis of the proposed CAR-MOSOA and correlated systems, for example, FDEAM [30], EOMR [31], TSGWO [32], and CoCoA [33], in the situation for consumption of energy, is presented in Figure 4. From Figure 4, it is indicated that a smaller amount of energy is consumed in the proposed CAR-MOSOA than that in the remaining three conventional methods. In the CAR-MOSOA method, fewer nodes are convoluted in the packet-promoting progression. Furthermore, a greater amount of energy is kept back as nodes through the maximum optimality factor that continually acquires significance to progress the data packets; quite the opposite, in conventional methods, an extra node is required in dispatching the equivalent data packet; accordingly, more energy is expended in the existing FDEAM [30], EOMR [31], TSGWO [32], and CoCoA [33] methods. 

Table 2 shows the comparative analysis of the energy consumption performance. The table shows that the proposed CAR-MOSOA achieves less consumption of energy when compared with existing techniques. In Figure 4, it witnesses that the CAR-MOSOA method has an outstanding reduction in the consumption of energy in comparison with the remaining three methods, as it exhibits consistency and consumes the smallest amount of energy to establish the ideal route from base to endpoint. Moreover, Figure 4 shows that consumption of energy rises as soon as node count rises in the setup.

### 6.2. Performance Based on End-to-End Delay

The performance analysis of end-to-end delay with CAR-MOSOA and existing methods (FDEAM [30], EOMR [31], TSGWO [32], and CoCoA [33]) is illustrated in Figure 5. From Figure 5, it can be seen that CAR-MOSOA achieves less delay and continually attempts to transfer the data packet to the finest intermediary node, which gratifies the optimality feature. Quite the opposite, the conventional FDEAM [30], EOMR [31], TSGWO [32], and CoCoA [33] take a considerable period to determine the endpoint node. Moreover, the reply message of the existing methods returns to the source node, only after determining the destination node. Therefore, the typical end-to-end delay in the associated system is longer than that in the CAR-MOSOA method. Table 3 shows the comparative analysis of end-to-end delay performance. From the table, it can be seen that the proposed CAR-MOSOA achieves less time delay when compared with existing techniques.

### 6.3. Performance Based on Packet Delivery Ratio

The performance analysis of packet delivery ratio with CAR-MOSOA and existing methods (FDEAM [30], EOMR [31], TSGWO [32], and CoCoA [33]) is illustrated in Figure 6. From Figure 6, it can be seen that CAR-MOSOA achieves better PDR in relation to the existing methods. Moreover, the ratio of PDR turns out to be high once the corresponding communication contains single and binary values. Table 4 shows the comparative analysis of the packet delivery ratio performance. From the table, it can be seen that the proposed CAR-MOSOA delivers more data packets at a specified node when compared with existing techniques.

### 6.4. Performance Based on Network Lifetime

The performance analysis of network lifetime with CAR-MOSOA and existing methods (FDEAM [30], EOMR [31], TSGWO [32], and CoCoA [33]) is illustrated in Figure 7. From Figure 7, we can see that CAR-MOSOA achieves better results in relation to the existing methods. In the existing FDEAM [30], EOMR [31], TSGWO [32], and CoCoA [33] methods, once node count rises in the network, additional sensor nodes begin to transfer the data packets indiscriminately. Moreover, the node can expire at a certain time period. In the CAR-MOSOA method, the ideal node is designated to transfer the packets, which causes a rise in battery life and network lifetime. Table 5 shows the comparative analysis of the packet delivery ratio performance. From Table 5, it is seen that the proposed CAR-MOSOA increases the network lifetime at a specified node when compared with existing techniques.

From all the analyses, the proposed CAR-MOSOA shows that better QoS in WSN is attained with a multi-tier framework. The CAR-MOSOA improves the performance of the network and its QoS by executing a routing procedure to transfer the packets from a base station to the endpoint by choosing the ideal route. The communication cost considered in the clustering and routing process is used to minimize the energy consumption of the network. The packet delivery is improved by generating the collision-aware routing path using the MOSOA. Since, an optimal collision-aware transmission path is discovered by considering the queue length, link quality, communication cost, and residual energy in MOSOA-based routing process. Moreover, the PDR is also improved by avoiding node failure based on the residual energy considered in the CH selection. Therefore, the developed CAR-MOSOA obtains a higher PDR while achieving a greater network lifetime in the WSN-based IoT.

## 7. Conclusions

Advancements in computer technology have contributed to the growth of WSNs, which sense the requisite parameters at any time. The WSN-based IoT systems are gaining huge attention in recent times. During point-to-point transmission, these systems suffer from restricted bandwidth, low power, and limited resources. Collecting the data is a familiar method for alleviating the above-mentioned problem, but the key problem in sensor networks is how to process the collected data efficiently. Therefore, in this paper, collision-aware routing using a multi-objective SOA (CAR-MOSOA) is proposed to transmit the alert messages using the IoT devices to the respective destination with a significantly reduced delay. The proposed works defining the role of IoT in WSN are analyzed and then the various data aggregation approaches are presented and compared with previous works. The data aggregation techniques focus on energy conservation, lifetime enhancement, better QoS, and high-level security of the network. The overall simulation clearly shows that the proposed CAR-MOSOA achieved better results by minimizing the energy consumption and end-to-end delay by up to 25% and 34%, respectively. Moreover, the network lifetime and PDR increased by up to 5.96% and 3.15%, respectively, when compared to the TSGWO method. In the future, this research will be extended to use hybrid optimization techniques to improve the performance of the network.

## Figures and Tables

**Figure 1 sensors-21-08496-f001:**
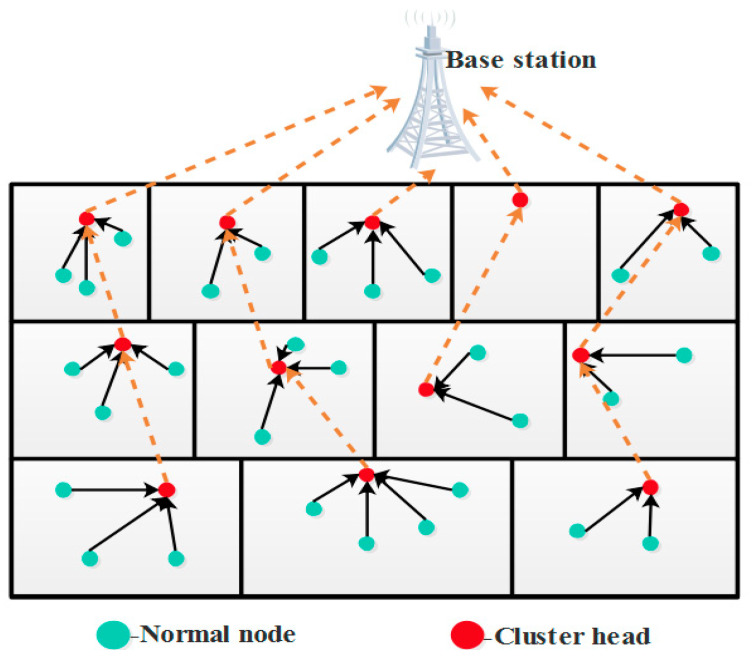
Proposed multi-tier framework.

**Figure 2 sensors-21-08496-f002:**
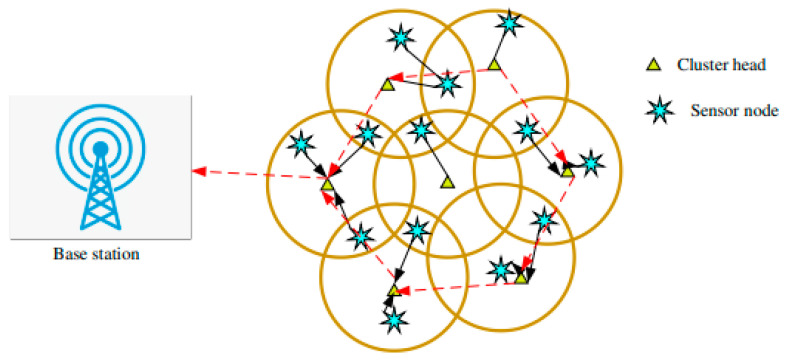
Network model of WSN-IoT.

**Figure 3 sensors-21-08496-f003:**
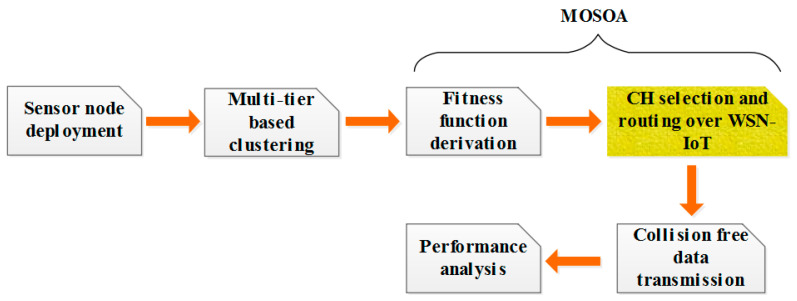
Overall block diagram of WSN-IoT.

**Figure 4 sensors-21-08496-f004:**
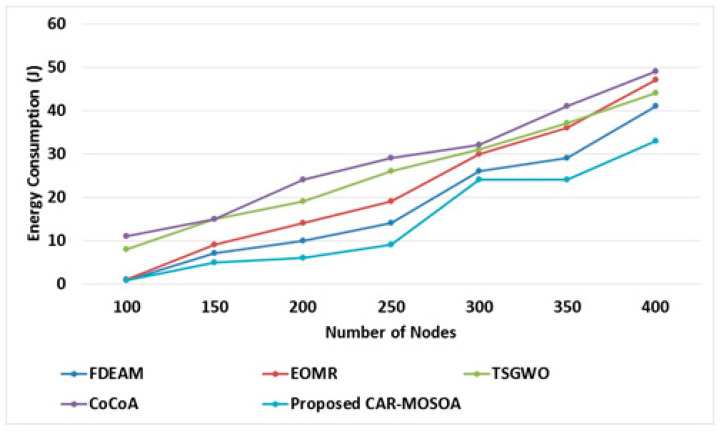
Performance based on energy consumption.

**Figure 5 sensors-21-08496-f005:**
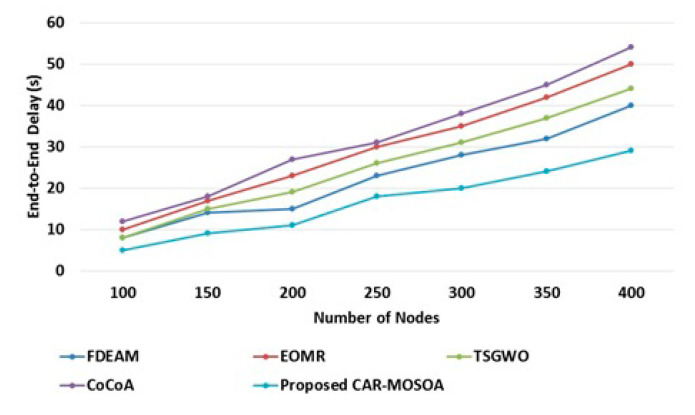
Performance based on end-to-end delay.

**Figure 6 sensors-21-08496-f006:**
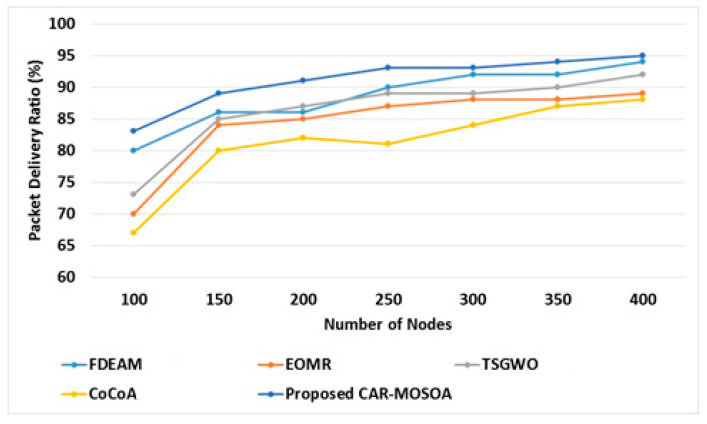
Performance based on packet delivery ratio.

**Figure 7 sensors-21-08496-f007:**
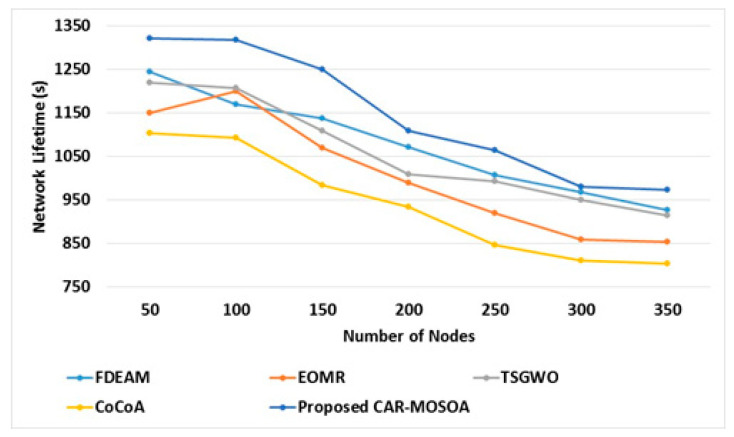
Performance based on network lifetime.

**Table 1 sensors-21-08496-t001:** Simulation parameters.

Constraints	Value
Transmission power (W)	1.4
Traffic type	CBR
Simulation time (s)	100
Receiving power (W)	1.0
Packet size (bytes)	512
Number of nodes	400
Network size (m)	500 × 500
MAC protocol	IEEE 802.15.4
Initial energy (J)	100
Data transfer rate (Kb/s)	250

**Table 2 sensors-21-08496-t002:** Comparative analysis of energy consumption.

Number of Nodes	Energy Consumption (J)				
	FDEAM [30]	EOMR [31]	TSGWO [32]	CoCoA [33]	Proposed CAR-MOSOA
100	0.8	1	8	11	0.8
150	7	9	15	15	5
200	10	14	19	24	6
250	14	19	26	29	9
300	26	30	31	32	24
350	29	36	37	41	24
400	41	47	44	49	33

**Table 3 sensors-21-08496-t003:** Comparative analysis of end-to-end delay.

Number of Nodes	End-to-End Delay (s)				
	FDEAM [30]	EOMR [31]	TSGWO [32]	CoCoA [33]	Proposed CAR-MOSOA
100	8	10	8	12	5
150	14	17	15	18	9
200	15	23	19	27	11
250	23	30	26	31	18
300	28	35	31	38	20
350	32	42	37	45	24
400	40	50	44	54	29

**Table 4 sensors-21-08496-t004:** Comparative analysis of packet delivery ratio.

Number of Nodes	Packet Delivery Ratio (%)				
	FDEAM [30]	EOMR [31]	TSGWO [32]	CoCoA [33]	Proposed CAR-MOSOA
100	80	70	73	67	83
150	86	84	85	80	89
200	86	85	87	82	91
250	90	87	89	81	93
300	92	88	89	84	93
350	92	88	90	87	94
400	94	89	92	88	95

**Table 5 sensors-21-08496-t005:** Comparative analysis of network lifetime.

Number of Nodes	Network Lifetime (s)				
	FDEAM [30]	EOMR [31]	TSGWO [32]	CoCoA [33]	Proposed CAR-MOSOA
50	1245	1150	1220	1103	1321
100	1170	1200	1208	1093	1319
150	1138	1070	1109	984	1251
200	1072	990	1010	935	1110
250	1007	920	994	847	1064
300	968	860	951	811	981
350	927	853	915	803	973

## Data Availability

No new data were created or analyzed in this study. Data sharing is not applicable to this article.

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
