# Peer review of "Collision-Aware Routing Using Multi-Objective Seagull Optimization Algorithm for WSN-Based IoT"

_sensors, 2021, doi:10.3390/s21248496_

Round 1
Reviewer 1 Report
The overview of the manuscript is quite well written. Presentation is clear. However, there is a lack of discussion of relevant research results that will help readers see its academic contributions more clearly.
This manuscript contains references to cutting-edge research. However, for completeness, they may also consider referencing this research paper.
Cao, Y., Li, Y., Jermsittiparsert, K., & Razmjooy, N. (2019). Experimental Modeling of PEM Fuel Cells Using a New Improved Seagull Optimization Algorithm. Energy Reports, 5, 1616-1625.
Author Response
Reviewer 1:
Comments and Suggestions for Authors
The overview of the manuscript is quite well written. Presentation is clear. However, there is a lack of discussion of relevant research results that will help readers see its academic contributions more clearly.
Answer:
Thank you for the useful comment. As per your comment, we have provided the discussion about the proposed research with better results at the end of section 6.
This manuscript contains references to cutting-edge research. However, for completeness, they may also consider referencing this research paper.
Cao, Y., Li, Y., Jermsittiparsert, K., & Razmjooy, N. (2019). Experimental Modeling of PEM Fuel Cells Using a New Improved Seagull Optimization Algorithm. Energy Reports, 5, 1616-1625.
Answer:
Thank you for the useful comment. Based on your comment, we have cited the aforementioned reference as Ref [42] in the manuscript.
Reviewer 2 Report
In this article, the authors develop collision aware routing using multi objective seagull optimization to reduce congestion in the network. The results show that the proposed method has better performance.
According to the above, clear goals have been set and, the authors made a focused presentation and analysis. Also, they managed to convey the research findings to the reader.
The title of the paper reflects its content. The article is well written, easy to follow and understand. This article includes lots of references to other relevant studies. Most of them are up-to-date. Although the paper is well written, the following suggestions will improve its quality:
(1) The results presented in Figs. 4-7 must have a better resolution because their current quality is moderate.
(2) There are some articles that talk about the congestion control mechanism in the CoAP, which is an application layer protocol devised for the IoT. These are listed below:
-- A. Betzler, C. Gomez, I. Demirkol and J. Paradells, "CoAP congestion control for the internet of things," IEEE Communications Magazine, vol. 54, no. 7, pp. 154-160, July 2016.
-- August Betzler, Carles Gomez, Ilker Demirkol, Josep Paradells, CoCoA+: An advanced congestion control mechanism for CoAP, Ad Hoc Networks, Volume 33, 2015, Pages 126-139.
-- Carlo Vallati, Francesca Righetti, Giacomo Tanganelli, Enzo Mingozzi, Giuseppe Anastasi, Analysis of the interplay between RPL and the congestion control strategies for CoAP, Ad Hoc Networks, Volume 109, 2020.
-- Angelo P. Castellani, Michele Rossi, Michele Zorzi, Back pressure congestion control for CoAP/6LoWPAN networks, Ad Hoc Networks, Volume 18, 2014, Pages 71-84.
I suggest you should also include a comparison of your proposed scheme with the congestion control mechanisms mentioned in one these articles that are based on standard application layer protocol.
Author Response
Reviewer 2:
Comments and Suggestions for Authors
In this article, the authors develop collision aware routing using multi objective seagull optimization to reduce congestion in the network. The results show that the proposed method has better performance.
According to the above, clear goals have been set and, the authors made a focused presentation and analysis. Also, they managed to convey the research findings to the reader.
Answer:
Thanks for your observation and valuable time.
The title of the paper reflects its content. The article is well written, easy to follow and understand. This article includes lots of references to other relevant studies. Most of them are up-to-date. Although the paper is well written, the following suggestions will improve its quality:
(1) The results presented in Figs. 4-7 must have a better resolution because their current quality is moderate.
Answer:
Thank you for the useful comment. Based on the results taken from the simulator, we have generated the results using Microsoft Excel to provide a good quality image in the manuscript.
(2) There are some articles that talk about the congestion control mechanism in the CoAP, which is an application layer protocol devised for the IoT. These are listed below:
-- A. Betzler, C. Gomez, I. Demirkol and J. Paradells, "CoAP congestion control for the internet of things," IEEE Communications Magazine, vol. 54, no. 7, pp. 154-160, July 2016.
-- August Betzler, Carles Gomez, Ilker Demirkol, Josep Paradells, CoCoA+: An advanced congestion control mechanism for CoAP, Ad Hoc Networks, Volume 33, 2015, Pages 126-139.
-- Carlo Vallati, Francesca Righetti, Giacomo Tanganelli, Enzo Mingozzi, Giuseppe Anastasi, Analysis of the interplay between RPL and the congestion control strategies for CoAP, Ad Hoc Networks, Volume 109, 2020.
-- Angelo P. Castellani, Michele Rossi, Michele Zorzi, Back pressure congestion control for CoAP/6LoWPAN networks, Ad Hoc Networks, Volume 18, 2014, Pages 71-84.
I suggest you should also include a comparison of your proposed scheme with the congestion control mechanisms mentioned in one these articles that are based on standard application layer protocol.
Answer:
Thank you for the useful comment. We have included the aforementioned references (i.e., Ref [6], [12], [13] and [33]) in the manuscript. As per your comment, we have used the Ref [33] entitled “CoAP congestion control for the internet of things” for comparison purposes.
Reviewer 3 Report
The autors present a new meta-heuristic optimisation algorithm (Seagull Optim Algo) to make clustering and routing
Surprisingly, the original MOSOA papers of 2019 and 2021 are not cited
Yet another clustering based routing protocol for WSN (with energy criteria)
The authors need to motivate why MOSOA is interesting in this context
[25][26] and [27] are not reference papers in valuable journals ([25] is not cited once..)

Author Response
Reviewer 3:
Comments and Suggestions for Authors
The autors present a new meta-heuristic optimisation algorithm (Seagull Optim Algo) to make clustering and routing
Surprisingly, the original MOSOA papers of 2019 and 2021 are not cited
Answer:
Thank you for the useful comment. Based on your comment, we have cited the original MOSOA papers of 2019 and 2021 as Ref [24] and [25] respectively.
Yet another clustering based routing protocol for WSN (with energy criteria)
The authors need to motivate why MOSOA is interesting in this context
Answer:
Thank you for the useful comment. As per your comment, we have included the reason for choosing the SOA in the introduction section.
[25][26] and [27] are not reference papers in valuable journals ([25] is not cited once..)
Answer:
Thank you for the useful comment. In the updated paper, the Ref [25], [26] and [27] becomes Ref [30], [31] and [32]. Based on your comment, we have verified the aforementioned references. These papers are from the journals of the Wiley Online Library, Springer and Tech Science Press which are indexed in SCIE, SCIE and SCI journals respectively. Moreover, the Ref [30] is cited in the section 2.